# The Current Status of Public Playground Safety and Children’s Risk Taking Behavior in the Park: Nakhon Si Thammarat Province, Thailand

**DOI:** 10.3390/children9071034

**Published:** 2022-07-12

**Authors:** Preeda Sansakorn, Uraiwan Madardam, Jutaluck Pongsricharoen, Narumon Srithep, Nisarat Janjamsri, Jittaporn Mongkonkansai

**Affiliations:** Department of Occupational Health and Safety, School of Public Health, Walailak University, Nakhon Si Thammarat 80000, Thailand; sansakorn_birmingham@hotmail.com (P.S.); muraiwan@wu.ac.th (U.M.); Jutaluck.apscr@gmail.com (J.P.); narumon.som38@gmail.com (N.S.); nisarat_2217@hotmail.com (N.J.)

**Keywords:** playground safety, play behavior, school children, equipment

## Abstract

The playground is perhaps the one area where school children feel like they can roam free, and public playgrounds provide many learning opportunities through different types of play. A cross-sectional descriptive study is presented with the objective of studying playground safety and the play behavior of primary school children at a park in Nakhon Si Thammarat Province, Thailand. The total number of playground equipment pieces was 22, the sample group of children was 362 children, and the data were collected using a playground safety survey and an observation form on playground play behavior. Data were collected from 2017 to 2018 and were analyzed using descriptive statistics including frequency, average, and standard deviation. The research showed that: (1) the most common defects of playground equipment included the material selection, the distance of the stair steps, and the height of the playground equipment; (2) the lack of awareness of children in terms of using the playground equipment safely; and (3) the three top risks in the play behavior of children were not checking equipment or toys before play, playing on the equipment over-adventurously, and playing on the equipment carelessly with friends. Related government agencies should provide support and management for playground areas and playground equipment by continuously implementing equipment checks, improvements, and repairs.

## 1. Introduction

Primary school children are at an important age where they should be supported and encouraged to have good development, whether physical, mental, intellectual, emotional, social or disciplinary, including the promotion of analytical thinking [1]. Learning can also take place outside of the classroom; it may be achieved in such places as playgrounds, botanical gardens, museums, aqua parks, zoos, rivers, and forests [2]. Play includes multiple phenomena; children’s experience of play is quite complicated, and may contain multiple dimensions of society, emotion, creativity, and imagination [3]. Thus, children can create their own play methods which go further than what teachers can do in a classroom. In 2012, children spent more time taking exams for academic knowledge than learning through play and research; playing and movement are very important during the life of a child [4]. The same result was obtained in 2018, when it was found that children also spent minimal time outdoors at school [5]. Permitting children to learn through play in the environment allows them to develop many skills, such as analytical thinking, social interaction, physical awareness, and evolved mental mechanisms [4,6,7]. Outdoor play for children motivates the development of children in all areas more than classroom play, and the playground environment or outdoor green areas increase their learning and health [8,9]. Children that are not provided the opportunity to play appropriately have a higher chance of being obese [10]. Therefore, the playground is a very important source of learning for young people.

A playground is a place that is especially designed for children to play, and it may be indoors but is usually outdoors [11]. Types of playground equipment can be divided into five groups: (1) trapeze equipment such as bars, loops, ladders, balance bars, parallel bars, etc.; (2) slip equipment such as slides, tube slides, etc.; (3) swing equipment such as swings; (4) motion equipment such as roundabouts, carousels, rocking horses, etc.; and (5) mixed equipment sets [9]. There are many benefits to the development of children through playing on playground equipment. However, there may be risks for children playing on playground equipment as well, such as inappropriate play behavior or unsafe playground equipment, which may cause accidents and injuries to occur. Fuselli et al. [12] found that there were approximately 29,000 children who were injured while playing on playground equipment in Canada in 2012, especially from falling from high places. In Thailand, children aged 1–14 years have died in playground accidents, and the mortality rate is calculated at 22/100,000. During 1999–2005, there was a high rate of death [13]. A study on play behavior in young children found that children use playground equipment incorrectly, for example, climbing up a slide from the bottom, scrambling to climb onto playground equipment, and swinging on playground equipment violently [14]. Sandseter [15] defines risky play for children as thrilling and exciting forms of play that involve a risk of physical injury. In addition to the behavior of children, playground equipment on a playground is an important factor contributing to injury in children. A study found that play area and playground equipment was improper and unsafe, for example, some playground equipment had safety rails, fences, and equipment heights that were not in line with safety standards [16,17]. If children go to play on such unsafe playground equipment as mentioned above, this will result in injuries.

The playground that was selected for this study is at the largest park with the most amount of playground equipment in Nakhon Si Thammarat Province. It is open every day, providing the opportunity for many children and young people to come and use this playground. Unsafe playground equipment and the risky play behavior of children, as mentioned, may cause accidents, injuries, or the loss of life. Factors affecting the risk of supervisory neglect (SN) and childhood injury are caregiver factors, child factors, and environmental characteristics [18]. Therefore, to ensure the safe supervision of the children’s play behavior and to enhance the development of children in many dimensions, the researcher was interested in studying the playground safety and play behavior of primary school children at this park in Nakhon Si Thammarat Province, Thailand.

## 2. Materials and Methods

This is a cross-sectional descriptive study. The total sample included 22 pieces of playground equipment at a park and primary school children who came to play at this playground. The sample consisted of all playground equipment at the playground (5 mixed playground equipment sets, 2 slides, 4 swings, 7 jungle gyms, 2 tube slides, 2 pieces of motion equipment) and focused on children aged 7–12 years old. Purposive sampling was used, which was based on the age of the children who came to play at the playground and their agreement along with parental consent to voluntarily participate in this research study. The total sample population was 362 people.

The materials and methods for this study consisted of the following:Playground safety survey—The applied survey adhered to the playground safety requirements of research coordination centers [9] to enhance safety and prevent injury to children. The survey consisted of 8 categories that included materials and selection, bolts and screws, the height and the gap size of the playground equipment, trapeze equipment, slip equipment, swing equipment, and motion equipment for a total of 52 survey items. The questionnaire’s reliability was 0.8.General questionnaire—The questionnaire consisted of 2 parts: general information and safety awareness. The general information was composed of 10 items which included general information queries on the following: gender, age, educational level, frequency of play on playground equipment, parental supervision, and impact from playing on the playground equipment. The safety awareness section for playing at the playground had 12 items on the questionnaire. The questions included both text and pictures. The 4 categories of questions were as follows: playground area, tube slide and slide, swings, and motion equipment. There were two types of safety awareness assessment criteria: correct and incorrect. The meaning of “correct” was to answer all questions correctly. An awareness assessment was incorrect if at least one of the answers was wrong. The questionnaire’s reliability was 0.9.Observation form—This form on the play behavior of children at the playground consisted of 13 noticeable issues with the evaluation of each issue as either risky behavior or not risky behavior. The questionnaire’s reliability was 0.9.

All questionnaires were tested for quality with the Index of Item Objective Congruence (IOC), which was equal to 0.8.

## 3. Quality Criteria for Measurement

The applied playground safety survey adhered to the playground safety requirements of research coordination centers [9]. The reliability of the playground safety survey was conducted by experts to check the accuracy of the tool. This survey has been used to explore other safe playgrounds. Before collecting the data, the researcher prepared the research team by training them to use the survey and trying out the survey together. As the research team who collected the data had a matching survey understanding, the questionnaire’s reliability was 0.8.

The general questionnaire and observation form were developed from a literature review via the following steps: (1) analyzing the research objectives to determine the variables to be studied; (2) researching the theory and literature on the variables to be investigated; (3) creating questionnaires and questions based on the research objectives; (4) finding the quality of accuracy by using 3 experts; (5) trying out the questionnaire with 30 children representing a sample of the province to determine the questionnaire’s confidence; (6) completing the questionnaire completely before using it in research. Therefore, this tool was of acceptable quality with an IOC greater than 0.5 and a Cronbach’s alpha reliability coefficient greater than 0.7.

## 4. Data Collection

Approval to conduct this research was granted by the Human Ethics Committee of Walailak University, Project Number: 16-142-01. Children were informed about the general details of the project and the presence of the observers prior to any data collection. Children were asked to take part in the usual playground activities and ignore the presence of the research staff in order to reduce the reactivity of the children.

Research staff visited the park to observe the children’s playground. These initial visits served the joint purpose of the research staff becoming familiar with the playground and the identified target areas and the children becoming familiar with the research staff.

Children were asked to complete questionnaires consisting of a set of safety awareness and general information questions as part of the data-gathering process. Each child’s questionnaire response took 10 min. The children were then able to play on several pieces of equipment after that. In this situation, the research staff observed how the children behaved in order to ensure whether or not the behavior was risky for the children involved. Each child played with each kind of playground equipment for 10 min. The same playground equipment had to be used at least five times in 10 min for each of the children. Each child’s playing behavior was watched for 60 min by the research staff.

## 5. Data Analysis and Statistics

Demographic data, playground equipment safety, safety awareness, and the play behavior of the children were evaluated by frequency, percentage, average, and standard deviation. Microsoft Excel was used for the analysis.

## 6. Results

### 6.1. General Information

Most of the sample group were female children (59.1%) with an average age of 8.9 ± 1.8 years who were studying at elementary level (62.7%). The average amount of times of play on the playground equipment at the park was 1.8 ± 0.9 times/week. For most of the play on the playground equipment, parents closely supervised their children, accounting for 63.0% of the play time. From the inquiries about the effects of play, it was found that six people in the sample group had accidents, accounting for 1.6% of the group. Bar or loop-type playground equipment accounted for 66.7% of the accidents. After the accidents, the sample groups had to take a break and then could continue to play again, including the sample group that had an awareness and knowledge about the proper use of the playground equipment (66.0%).

### 6.2. Playground Safety

It was found that the ground condition of the playground consisted of sand that was deeper than 20 cm, gutters around the playground, a falling area, and a standardized distance of free space and travel areas in accordance with playground safety inspection requirements. There were 22 pieces of playground equipment consisting of 5 mixed playground equipment sets, 2 slides, 4 swings, 7 jungle gyms, 2 tube slides, and 2 pieces of motion equipment. Inspection results found that the safety standard of the playground equipment failed at 100%. This means that the 22 items of playground equipment failed to meet the assessment criteria in all categories of equipment safety. The most common defects of the playground equipment included the material selection, the distance of the stair steps, and the height of the playground equipment (90.9%, 50.0%, and 33.3%, respectively). The details are shown in Table 1.

Considering the details of each piece of playground equipment, it was found that most of the playground equipment was damaged in condition, rusty, and lacked continuous maintenance, and thus did not comply with safety standards. The details are shown in Figure 1, Figure 2, Figure 3 and Figure 4.

### 6.3. Safety Awareness of Playgrounds and Playground Equipment

It was found that 80.0% of the children were not aware of the safety requirements of the playground and playground equipment. It was also found that children had the least accurate safety awareness for the following playground equipment: (1) the tube slide and slide, which accounted for 7.2%; and (2) the swing, which accounted for 9.9%. The details are shown in Table 2.

### 6.4. Play Behavior

It was found that the three top risky play behaviors of children on the playground equipment included not checking equipment or toys before play, playing on equipment over-adventurously, and playing on equipment carelessly with friends (92.5%, 81.0%, and 59.7%, respectively). The details are shown in Table 3.

## 7. Discussion

This study found that the ground conditions of the surrounding area were up to standard in accordance with the safety of playgrounds. However, the safety standard of the playground equipment failed at 100%. The most common defects of the playground were the material selection, the distance of the stair steps, and the height of the playground equipment (90.9%, 50.0%, and 33.3%, respectively). Unsuitable and unsafe playground equipment included equipment with no safety rails, equipment with height not to standard, and equipment in unsafe areas [15,16]. In the inspection, it was found that the material used for most of the playground equipment was metal which was old, rusty, and damaged in condition. Bare metal was used for platforms, slides, and steps; this equipment when exposed to direct sunlight may reach temperatures high enough to cause serious injuries from contact burns in a matter of seconds [19]. From surveying the stair steps, it was found that the vertical distance between the steps was more than 32 cm; the standard for step platforms in Thailand for children aged 5–12 years should be in the range of 23–32 cm. In addition, the open gaps between the steps were between 9 and 23 cm, which is a risk factor for children getting their heads stuck. For international step standards, the requirement is equal to 18 inches or about 45 cm for school-aged children, and if there are more step platforms than what is standard, the gaps must be closed [19]. The result of the study was that the height of the playground equipment failed to meet safety standards. The distance from the ground to the level of the playground equipment was over 1.8 m, and the playing area above the ground at a distance greater than 75 cm did not have safety guardrails. The findings are in line with research among children in the USA, which found that 34% of playground equipment for climbing did not meet safety standards and the playgrounds did not have guardrails or walls to protect against falls [19,20], which can lead to injury. Playground-related traumatic brain injuries are the result of children falling while playing at a playground, which may lead to brain injuries, skull fractures, or upper extremity injuries [12,21]. However, if the ground and playground equipment are improved in accordance with safety standards, this can reduce the injury rate of children. If the ground of a playground is changed from concrete to a mixture of wood and the height of the playground is reduced to 1.5 m, this can reduce the injury rate by 30%, compared to before the improvements are made and before new playground equipment is added [22].

Playgrounds and playground equipment can be risky and dangerous. Children can either choose to play or not play on risky playground equipment. However, children cannot avoid the hidden hazards of playing on a playground, such as risky slides that are too high or slip rails that are overheated [23]. In Thailand, there is a set of documented instructions on the proper use of label-controlled products for all equipment. However, installation manuals should be shown on the playground, and the safety condition of the playground equipment should always be checked by expert technicians regularly [23] or should follow the Canadian Requirements Association’s standards for maintaining playground equipment. This standard includes suggestions for technical specifications and procedures that should be followed when designing, manufacturing, constructing, installing, maintaining, and inspecting playground equipment and play areas for the general public [24] and refers to the US Consumer Product Safety Commission [19]. The requirements outlined in this standard are meant to reduce the possibility of severe and/or life-threatening injuries.

It was found in this study that, for the most part, children were not properly aware of how to safely use playgrounds and playground equipment. This was different from the study of Little and Eager [25], which found that children can be aware of which types of play are risky, complicated, and unsafe. Consequently, children who may be aware that the playground equipment is safe may also be willing to take greater risks [26]. However, school playground safety policies, safety rules, access to equipment, access to playground areas, and various levels of supervision within school playgrounds are important safety precautions to regulate the play behavior of children [27].

The three top risky play behaviors of children on playgrounds included not checking the equipment or playground equipment before play, playing on the equipment over-adventurously, and playing on the equipment carelessly with friends. In accordance with other research, it was found that children will choose the most challenging playground equipment, especially equipment that involves swinging, spinning, and climbing [14]. Children aged 6–12 years are considered school-aged children; behavior during this age is based on trial and error, observing the reactions of those around them, and needing acceptance from those around them, whether friends or family [28,29]. Therefore, children try to play in different ways in order to learn new things. If they receive attention from those around them, they may become more confident, resulting in risky behavior. Risky play is defined as: “thrilling and exciting forms of physical play that involve uncertainty and a risk of physical injury [30]. Children’s risk-taking and risky play have been linked with possible advantages in children’s learning and development. Research reveals that risky play can lead to increased physical activity, improved motor/physical competence, and a higher ability to assess risks and handle risk situations in an appropriate way, as well as general health and positive psychological outcomes [31,32,33,34]. Furthermore, it has been found that children’s desire to take risks is higher than what their mothers expect [35]. Moreover, children at school, in the absence of their parents but under the supervision of another adult, behaved according to their own wishes. There is a strong relation between children’s participation in daily activities, engaging with family and friends, and their subjective well-being [36].

This research has limitations in terms of children’s play because the playing time was limited and the children knew that the researchers were observing their play. This might have made them play unnaturally.

## 8. Conclusions

The core objective of this study was to study the playground safety and playground behavior of school children in a public playground. It was found that the children were not aware of the safety requirements of the playground and playground equipment. In addition, children have play behaviors that put them at risk of injury. As a result, parents, teachers and relevant local authorities should pay close attention to play and provide a safe play environment for children. It is important to examine equipment before children enter the play structure, and to keep an eye out for any issues during a play period.

## 9. Suggestions

Parents should check playground equipment for any sharp or dangerous surfaces and teach children playground safety rules and safe behavior;Teachers should teach specific rules that apply to the use of playground equipment and teach children to have concern for others in the school playground. In addition, school administrators should be held responsible for supervising children’s play at all times;Local authorities should inspect and maintain public playground equipment so that it is safe and ready for use. They should clearly identify a responsible person as well as allocate a budget for regular maintenance. Proper playground care involves regular inspections and updates of playground equipment and grounds. Developing a customized playground care and maintenance plan will help your playground provide the best possible play experience;A systematic collaboration between parents, teachers, directors, local administrators, and other related personnel with the owners of playgrounds and playground equipment should be established to ensure that the safety of the playground and playground equipment is sustainable;The roles of parents, parents’ safety education, local authorities, and material types that are appropriate—particularly for outdoor playground equipment in a humid tropical climate—could be the focus of further research.

## Figures and Tables

**Figure 1 children-09-01034-f001:**
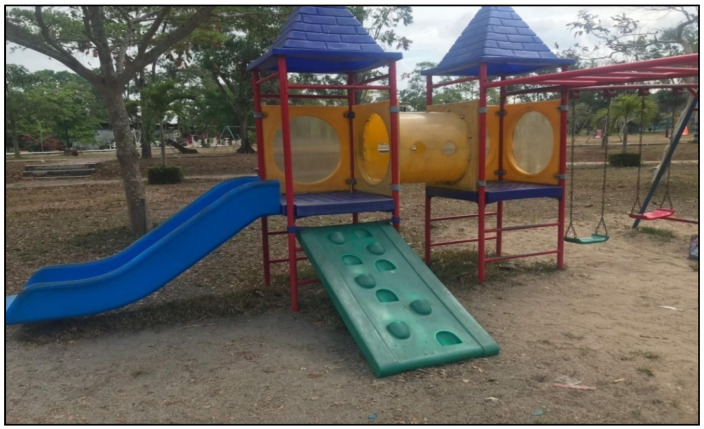
Mixed playground equipment set in the unsafe playground: arranged near other playground equipment; path for running had obstacles blocking the flow of traffic; damaged playground equipment; broken plastic parts.

**Figure 2 children-09-01034-f002:**
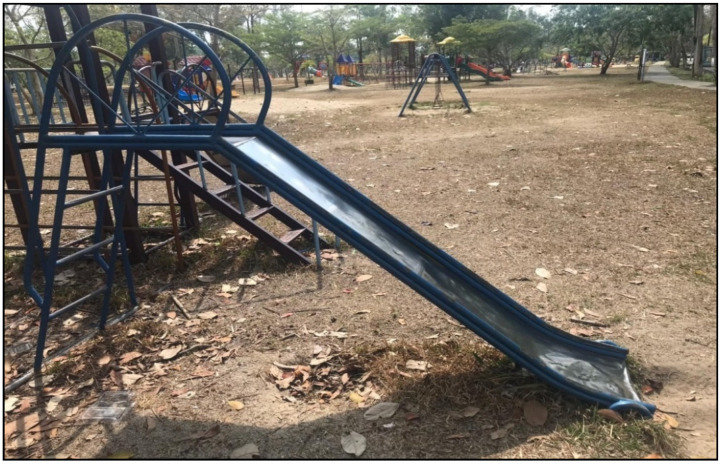
Slide set: There was no elevated area and no space to stand for some slides. The slide surface was rough and rusty. The sides of the slide had raised edges less than 10 cm high. There was no exit that was parallel to the floor. The length of the exit was not related to the length of the slide and the height between the floors to the exit.

**Figure 3 children-09-01034-f003:**
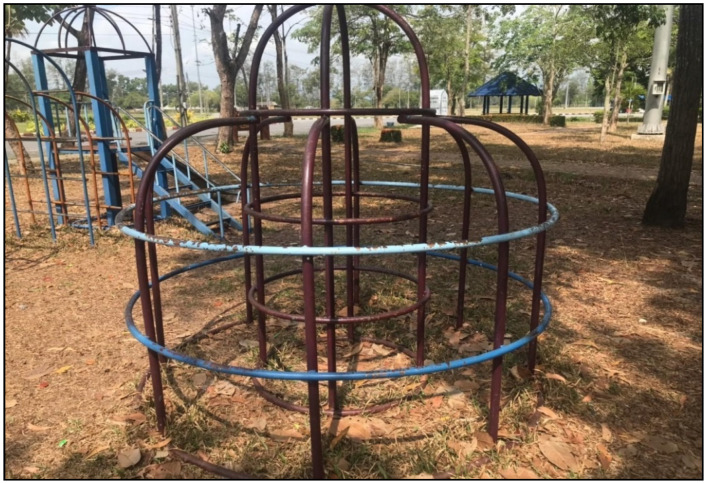
Jungle gym: The height from the floor was more than 1.8 m where a child might fall. There were no safety rails. The distance between the stairs was more than 32 cm. Color paint had peeled off of the playground equipment.

**Figure 4 children-09-01034-f004:**
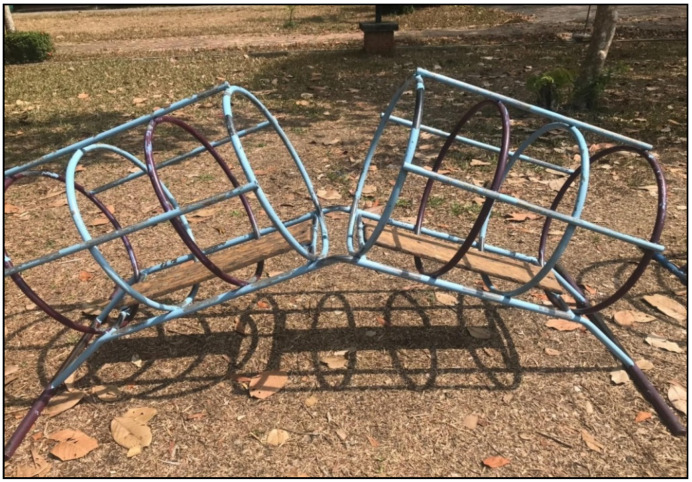
Tube slide: The diameter of the tube slide was less than 75 cm. Color paint had peeled off of the playground equipment.

**Table 1 children-09-01034-t001:** Playground safety at a park in Nakhon Si Thammarat Province, Thailand.

Defects	Qty (Pieces)	Passed the Criteria Amount (%)	Did Not Pass the Criteria Amount (%)
- Height of playground equipment	15	10 (66.7)	5 (33.3)
- Gap size of equipment	9	9 (100.0)	0 (0.0)
- Size of handle parts	11	11 (100.0)	0 (0.0)
- Vertical step distance	8	4 (50.0)	4 (50.0)
- Material selection	22	2 (9.1)	20 (90.9)

**Table 2 children-09-01034-t002:** Safety awareness of playground and playground equipment (*n* = 362).

Safety Awareness	Correct (%)	Incorrect (%)
1. Playground area	69 (19.1)	293 (80.9)
2. Tube slide and slide	26 (7.2)	336 (92.8)
3. Swing	36 (9.9)	326 (90.1)
4. Motion equipment	51 (14.1)	311 (85.9)

**Table 3 children-09-01034-t003:** Play behavior (*n* = 362).

Play Behaviors	Risk (%)	No Risk (%)
Article 1: Playing on damaged playground equipment	18 (5.0)	344 (95.0)
Article 2: Teasing each other while playing	97 (26.8)	265 (73.2)
Article 3: Playing around dangerous equipment	33 (9.1)	329 (90.9)
Article 4: Playing over-adventurously	293 (81.0)	69 (19.0)
Article 5: Playing correctly	22 (6.1)	340 (93.9)
Article 6: Picking dangerous objects to play with at the playground	88 (24.3)	274 (75.7)
Article 7: Mimicking the playing behaviors of friends	128 (35.4)	234 (64.6)
Article 8: Not checking equipment or toys before play	335 (92.5)	27 (7.5)
Article 9: Not being careful with friends while playing	216 (59.7)	146 (40.3)
Article 10: Complying with playground regulations	22 (6.1)	340 (93.9)
Article 11: Playing with equipment alone	138 (38.1)	224 (61.9)
Article 12: Wearing shoes while playing	76 (21.0)	286 (79.0)
Article 13: Using playground equipment suitable for age	108 (29.8)	254 (70.2)

## Data Availability

The datasets used and/or analyzed during the current study are available from the corresponding author on reasonable request.

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
