# Peer review of "The Current Status of Public Playground Safety and Children’s Risk Taking Behavior in the Park: Nakhon Si Thammarat Province, Thailand"

_children, 2022, doi:10.3390/children9071034_

Round 1

Reviewer 1 Report

I feel that this is an informative manuscript that has some merit in today's climate of safety in playgrounds. I suggest the authors look into 'risky play' from Ellen Sandseter for more reference on this topic. I do agree that this paper covers the safety concerns of the equipment but evaluating it from a child's perspective could yield different results. Risk-taking is valuable for a child to develop and grow. This could be teased out in the discussion section. However, this article is well-written and this is just a suggestion. 

Reviewer 2 Report

Thank you for your important, worthwhile research. This could make a valuable contribution to child safety in public places within Thailand, but much mor detail on your research process is required – please see my suggested changes below:

Please provide a reference to support the following statement:

‘Outdoor activity play provides many opportunities for children because it allows them to play independently and learn things quickly, which cannot be met in the classroom’

Reference needed:

‘Children these days spend more time taking exams for academic knowledge than learning through play and research, where playing and movement are very important during the life of a child’

Avoid the use of etc. – say what things you mean:

‘Permitting children to learn through play in the environment allows them to develop many skills, such as analytical thinking, social interaction, physical awareness, etc [2,3]’

Avoid saying abroad – be specific:

‘A study abroad found that there were approximately 29,000 children who have been injured while playing on playground equipment, especially falling from high places [9]’

Line 55 – please clarify what you mean by a violent accident. Please also specify the time period.

Is violent behaviour related to playground safety equipment?

During 1999-2005, there was a higher rate of death from accidents and violent behavior on playgrounds than in other years

Lines 71-76 are findings and should not be included in the introduction:

Lines 77-79 – are these findings, these concepts should be introduced in the introduction.

The numbers in the following paragraph do not correlate – please correct or explain clearly:

‘The total population included 22 play- ground equipment at a park and the sample included primary school children who came to play at this playground. The sample consisted of all playground equipment at the play- ground (9 bar equipment, 11 slip equipment, 4 swing equipment, and 2 motion equip- ment)’

Provide more information on each of the data collection tools used – who were they developed by, are they valid and reliable?

2.1 Data Collection – relates to ethical approval, please change

More information on recruitment, how participants were approached, who consented; is necessary. Also some discussion on the fact the participants knew they were being observed – what impact did that have?

Please rephrase the following sentence (110-111), I am not sure what the meanings is:

‘where data was collected by clarifying the objectives and providing details of the query data to the sample group of children and their parent’

More information on statistical analysis is needed – frequency and percentage of what etc. Was any software used to calculate?

How long did observations last for, how frequent, at what time of day, what season – more information is needed. Could you provide an observation schedule?

The numbers seem to be a smaller font, please rectify

Who completed the questionnaires? Parents/Researchers/Children – need to be much clearer.

Line 135 - What do you mean by – Failed at 100% - and how do the following percentages and Table 1, relate to it ((90.9%, 50.0%, and 33.3%, respectively)

Line 143 - How was ‘lacking in continuous maintenance’ assessed

Line 149 – The equipment at this playground will be removed and replaced.’ Please explain if this is a recommendation and who will remove and replace it.

Same comment for Line 154 – is this a recommendation?

Could you provide more information on how safety awareness was assessed?

Please explain the term J- standard height – line 200

Again avoid saying studies abroad and foreign research – be specific

Would be useful to have some discussion on the benefits of adventurous, risky play and a discussion on risk tolerance

Lines 248-253 seem to contradict each other

Line 258-259 – by who?

References are inappropriately formatted – no need to repeat the reference number.

Reference number 1 is not formatted correctly

Reviewer 3 Report

Authors of material research took into consideration a very important problem of public playground safety level and risk behavior among children.

Authors indicated that the safety standard of the playground equipment failed at 100% and the lack of awareness of children in using the playground equipment safety.

I would recommend taking into consideration role of parents (safety education) and local authorities in the future research materials.

Material and methods were used appropriate to the research aims and problems.

Conclusions that authors indicated also present significiant data.

In discussion authors corresponded with the research results adequately to the problem though I would recommend refer also to European authors and instiitutional safety regulations.

Correct number and quality of references used in material research.

Round 2

Reviewer 2 Report

Thank you for making the amendments, they have certainly improved the manuscript. However, I still think the manuscript requires further editing before it is ready to be published. 

I think that 'fail at 100%' needs to be explained in the manuscript and should be removed from the abstract, or written in a way in which it is easy to understand what is meant by it.

Line 39 - remove the words 'these days' and replace with the time period in question, now and the time you are comparing against.

Line 55-56 - please specify the time period

Line 57-58 - please rephrase, to say high rates of death, rather than playground of death, no parents would take their child there.

Line 90 - please provide reference for the questionnaires reliability score, or explain how it was calculated - was the survey piloted before the study, if so explain this, was any training on how to do the observations given? If so say so. It is very important to understand how the data was collected. 

Line 92, the number '10' appears to be in a smaller font than the rest of the words

Line 91-99 - can you please explain if you developed the questionnaire, or if it is used by other researchers - if so include a reference. If you developed the questionnaire, can you describe the process briefly, and say whether it was piloted? Also comment on how reliability was calculated.

Line 100 - Please also explain if this is an already valid and reliable tool used in research/practice, or if you have developed it?

Line 118-120 - so the children were instructed which equipment and how long to play on them for? This is arguably not them playing in their natural environment - children may naturally choose to avoid certain equipment because they do not know how to use it, but if the research team has told them to spend 10 minutes on the equipment, that could skew the data to more unsafe practices. Can you please clarify that my interpretation is correct, and if so, can you acknowledge as a limitation of the research.

Can you explain if you noticed that the children were putting themselves or others in danger, did you intervene?

Line 124 - please provide extra information, e.g. frequencies, percentages and standard deviations of the number of children using equipment unsafely was calculated etc...

Line 125 - please change to Microsoft excel was used for analysis

Please be consistent with number of decimal places used throughout

Please explain in the manuscript what fail at 100% means - does it mean that all equipment failed every item on the checklist, at the moment it is still not clear.

You have no table 2 - please update all table numbers and text

Line 267 - no need to include author initials in citation- change throughout

Line 286 - please add year for Howard et al study

Line 295-300 - can you clarify if you are saying Thailand do, or should, follow the Canadian guidelines

Line 341-345 appear to be formatted differently to the rest of the paper

Reference list needs to be numbered and must correspond with numbers in text. 
